# Timing-Attack-Resistant Acceleration of NTRU Round 3 Encryption on Resource-Constrained Embedded Systems

Eros Camacho-Ruiz * , Macarena C. Martínez-Rodríguez , Santiago Sánchez-Solano and Piedad Brox

Instituto de Microelectrónica de Sevilla, IMSE-CNM, CSIC/University of Seville, 41092 Seville, Spain
* Correspondence: camacho@imse-cnm.csic.es

**Abstract:** The advent of quantum computing with high processing capabilities will enable brute force attacks in short periods of time, threatening current secure communication channels. To mitigate this situation, post-quantum cryptography (PQC) algorithms have emerged. Among the algorithms evaluated by NIST in the third round of its PQC contest was the NTRU cryptosystem. The main drawback of this algorithm is the enormous amount of time required for the multiplication of polynomials in both the encryption and decryption processes. Therefore, the strategy of speeding up this algorithm using hardware/software co-design techniques where this operation is executed on specific hardware arises. Using these techniques, this work focuses on the acceleration of polynomial multiplication in the encryption process for resource-constrained devices. For this purpose, several hardware multiplications are analyzed following different strategies, taking into account the fact that there are no possible timing information leaks and that the available resources are optimized as much as possible. The designed multiplier is encapsulated as a fully reusable and parametrizable IP module with standard AXI4-Stream interconnection buses, which makes it easy to integrate into embedded systems implemented on programmable devices from different manufacturers. Depending on the resource constraints imposed, accelerations of up to 30–45 times with respect to the software-level multiplication runtime can be achieved using dedicated hardware, with a device occupancy of around 5%.

**Keywords:** hardware security; post-quantum cryptography; NTRU; embedded systems; resource-constrained devices; IoT

## 1. Introduction

The security of most digital infrastructure relies on public key cryptography (PKC), which enables secure communications between entities without sharing any pre-established secret. PKC provides (i) protected channel establishment (key establishment) and (ii) authentication of digital information (including authentication of individuals involved in a communication protocol through the application of digital signatures). The strength of current PKC techniques is based on the computation complexity of two mathematical problems: the factorization of large numbers and the computation of discrete logarithms. However, although these problems are complex for current state-of-the-art systems with high amounts of resources and computational power, they can be solved in a reasonable amount of time using quantum computers. As a consequence, the security of cryptographic protocols applied in our everyday life will be compromised in the near future. For instance, Shor's algorithm [1] highlights the capability of quantum computers in efficiently factoring integers. This exposes a weakness in the widely used RSA algorithm, which relies on the complexity of factoring a large biprime number. Additionally, Shor's algorithm can also solve the discrete logarithm problem (DLP) in polynomial time. The DLP serves as the foundation for other cryptographic methods, such as Diffie–Hellman (DH), the digital signature algorithm (DSA), and elliptic curve cryptography (ECC).

The scientific community has developed post-quantum cryptography (PQC) to deal with this threat. The roadmap of the EU Cybersecurity Strategy identifies PQC as a key enabling technology, as reported by ENISA (European Union Agency for Cybersecurity) in [2]. Moreover, the NIST (National Institute of Standards and Technology) started a post-quantum cryptography competition in 2016 to identify cryptographic algorithms able to withstand quantum computer attacks by 2022, the year in which the first algorithms to be standardized were presented [3]. Proposals submitted to the NIST PQC contest included software implementations. However, the design of hardware-efficient solutions is an open challenge for the electronics engineering community. Recent studies present the use of hybrid hardware/software (HW/SW) co-design methodologies to combine flexibility and efficiency when implementing PQC-based algorithms [4,5].

Among lattice-based PQC cryptosystems, the public key encryption scheme NTRU (Nth-degree truncated polynomial ring unit) was consolidated as a reference since it offers certain advantages over other cryptosystems with the same security level, namely, that it is faster and works with smaller key sizes [6]. NTRU's security is based on the shortest vector problem (SVP), which is a difficult problem in lattice reduction. Until now, no algorithm has been developed to solve this problem in polynomial time. The NTRU public key cryptosystem was standardized by the Institute of Electrical and Electronics Engineers (IEEE) in 2008 as IEEE Std 1363.1-2008 [7] and by the American National Standards Institute (ANSI) in 2010 as ANSI Std X9.98 [8]. The original version of NTRU has been progressively improved to be resilient against different types of attacks. NTRUEncrypt [9] and NTRU-HRSS-KEM [10] submissions in Round 1 of the NIST PQC standardization contest were merged in Round 2 to give rise to a new NTRU submission (NTRU [11]), which reached Round 3. The first list of PQ algorithms to be standardized was recently announced, on which NTRU is not among those selected. However, NTRU-based algorithms are a fundamental pillar in PQC with a solid background. Advances to provide efficient NTRU implementations on embedded systems are an open challenge, especially in certain scenarios where strict restrictions make the adoption of other PQC finalists with higher levels of complexity unfeasible.

There are a wide variety of implementations of NTRU encryption and decryption schemes on several platforms, such as software on embedded microcontrollers [12], field-programmable gate arrays (FPGAs) [13], and even an experimental study of hardware-dedicated building blocks for VLSI integrations [14]. In most cases, these implementations must be included in IoT environments where area and time constraints are very limited. This is because the evolution of programmable devices has progressed towards systems-on-chips (SoCs), which combine one or more embedded processor cores and programmable logic, encouraging the development of hybrid implementations following HW/SW co-design methodologies. The idea behind HW/SW implementations is to exploit the flexibility coming from software with the efficiency of hardware realizations for the most demanded timing operations. In the NTRU cryptosystem, the critical operation is the multiplication in the nth-degree truncated polynomial ring; thus, the efforts of the scientific community have been focused on its acceleration through hardware implementations. Most of the studies reported in the literature follow two well-distinguished methodologies. On one hand, some studies are based on a high-level synthesis (HLS) methodology, starting from a high-level description of the NTRU algorithm [15]. On the other hand, some employ a methodology based on a register-transfer level (RTL) description for critical operations [16–18]. The main advantage of the first strategy is the reduction in development time due to the use of automatic synthesis tools that do not require a solid background of designers in hardware description languages. However, the second strategy generally offers the most efficient implementations in terms of timing, power consumption, and area, using ad hoc hardware realizations for critical operations.

This paper presents the implementation of the finalist NTRU version included in the third round of the NIST PQC contest, following a HW/SW co-design methodology. Additionally, despite the efficiency of post-quantum cryptography, implementations of

lattice-based cryptography secure against side-channel attacks remain an open issue, as [19] and [20] point out. The security implementation aspect of lattice-based cryptography has yet to be explored in this regard. Some advances related to timing attacks are included in [21]. Therefore, this work (unlike others presented on NTRU) tries to include a solution that can involve a mitigation against timing side-channel attacks. Moreover, this implementation follows a flexible design that can mitigate timing-based side-channel attacks and also make it suitable for the area or temporal limitations that are common in IoT environments, establishing a compromise between area and performance. The selected platform is a modern SoC that includes a general-purpose processor and programmable logic, where an ad hoc serial multiplier will be implemented to accelerate the polynomial multiplications. The main contributions of this work are as follows:

- A specific solution for the NTRU polynomial multiplier in resource-constrained devices, where the availability of resources and the energy budget are very limited (which is common in the IoT integration framework), which allows for acceleration without generating any security breaches related to timing attacks in the system.
- The design of a highly configurable intellectual property (IP) module to implement an ad hoc serial polynomial multiplier on the programmable logic included in the SoC. The configuration enables the possibility to easily implement different security parameters defined in the NTRU algorithm scheme, as well as different arithmetic units responsible for performing the multiplication operation.
- The design of an interconnection scheme based in an AXI4-Stream protocol that optimizes the bandwidth of communication infrastructures between the processor core and the IP.
- The evaluation of (i) the resources used for each particular solution and comparison with other implementations in the literature; and (ii) the acceleration factors achieved with the proposed implementations using the software implementation of the NTRU algorithm included in the third round of the NIST PQC contest as reference. In addition, a figure of merit called *Efficiency* is proposed, which enables one to know which implementation achieves the best trade-off between a high acceleration factor and a moderated value of power consumption and area occupation.

The organization of this paper starts with Section 2, where the mathematical background is presented and a simple implementation of the polynomial multiplication at hardware level is illustrated. Section 3 introduces a method to accelerate the multiplication process that avoids the possibility of applying timing attacks to the simple implementation. Section 4 specifies the core design of the IP module and the embedded system integration. Section 5 presents the results in terms of resource consumption, timing performance, and efficiency of the IP module operating in the NTRU Round 3 version. Finally, the main conclusions of the work are summarized in Section 6.

## 2. The NTRU Encryption Scheme

### 2.1. Mathematical Background

The key encapsulation mechanism (KEM) for NTRU [11] is inherited from the NTRU-HRSS-KEM version submitted to the first round of the NIST PQC contest, which was based on a variant of the Fujisaki–Okamoto transformation [22]. Because the NTRU Round 3 version merges from the NTRUEncrypt and NTRU-HRSS-KEM cryptographic algorithms presented in the second round of the NIST PQC contest, two complementary variants are combined in the cryptosystem sample space: NTRU-HPS (Hoffstein, Pipher, and Silverman) from NTRUEncrypt [23] and NTRU-HRSS (Hülsing, Rijnveld, Schanck, and Schwabe) from NTRU-HRSS-KEM [10].

The cryptography scheme of NTRU is based on polynomial convolution rings or quotient rings, which are a particular algebraic structure where polynomial operations are performed [23]. The characteristics of each quotient ring are set depending on the NTRU Round 3 security level, which is modulated by the sets of parameters defined in [11]. The polynomial degree is configured by the parameter $n$, and the modulus of the

polynomial coefficients is set by the parameter $q$. In this scope, any polynomial whose coefficients are integers is denoted as $\mathbb{Z}[x]$.

One of the most important parts of the KEM is the encryption, whose scheme in NTRU Round 3 [11] requires defining the quotient rings described by (1)–(3):

$$R/q = \mathbb{Z}[x] \ / \ (q, \Phi_1 \Phi_n) \tag{1}$$

$$S/q = \mathbb{Z}[x] \ / \ (q, \Phi_n) \tag{2}$$

$$S/3 = \mathbb{Z}[x] \ / \ (3, \Phi_n) \tag{3}$$

where

- $\Phi_1$ is the polynomial $(x - 1)$;
- $\Phi_n$ is the polynomial $(x^n - 1)/(x - 1) = x^{n-1} + x^{n-2} + \ldots + 1$;
- $(q, \Phi_1 \Phi_n)$ represents the operation modulus $q$ for the coefficients and modulus $\Phi_1 \Phi_n$ for the polynomial degree;
- $(q, \Phi_n)$ represents the operation modulus $q$ for the coefficients and modulus $\Phi_n$ for the polynomial degree;
- $(3, \Phi_n)$ represents the operation modulus 3 for the coefficients and modulus $\Phi_n$ for the polynomial degree.

Thus, Equations (1)–(3) can be expressed as shown in (4)–(6).

$$R/q = \mathbb{Z}[x] \mod (q, x^n - 1) \tag{4}$$

$$S/q = \mathbb{Z}[x] \mod \left( q, \frac{x^n - 1}{x - 1} \right) \tag{5}$$

$$S/3 = \mathbb{Z}[x] \mod \left( 3, \frac{x^n - 1}{x - 1} \right) \tag{6}$$

Therefore, $R/q$ is a polynomial with a degree of $n - 1$ at most, with coefficients of $\{-q/2, -q/2 + 1, \ldots, q/2 - 1\}$, whereas $S/q$ is a polynomial with a degree of $n - 2$ at most, with coefficients of $\{-q/2, -q/2 + 1, \ldots, q/2 - 1\}$, and $S/3$ is a polynomial with a degree of $n - 2$ at most, with coefficients of $\{-1, 0, 1\}$, which constitutes the so-called *ternary* polynomial, $\mathcal{T}$. In this version of the NTRU, it is necessary to define a subset of ternary polynomials, represented by $\mathcal{T}(t)$, that contain exactly $t/2$ elements equal to $+1$ and other $t/2$ elements equal to $-1$.

The NTRU Round 3 encryption scheme initially requires two polynomials: the first one, which emerges from the public key $h(x) \in S/q$, and the blinding polynomial, which is a ternary polynomial, $r(x) \in \mathcal{T}$. Unlike previous versions of the NTRU cryptosystem, the number of nonzero coefficients of $r(x)$ is not known in this version. These two polynomials are multiplied according to the convolution product described by (7):

$$e(x) = r(x) \times h(x) \mod (q, x^n - 1) \tag{7}$$

where $e(x) \in R/q$. On the other hand, the message, which is the other polynomial required, is transformed into a ternary polynomial with $q/16 - 1$ ones and $q/16 - 1$ minus-ones, $m(x) \in \mathcal{T}(q/8 - 2)$, to increase the message obfuscation in the encryption process. Unlike NTRUEncrypt, the padding mechanism of the message presented in [10] disappears. However, a change in the message representation from $S/3$ to $R/q$ is required for NTRU Round 3. In NTRU-HPS, the new message $m'(x) \in S/3$ is equivalent to the message $m(x)$. In NTRU-HRSS, the message $m'(x)$ is obtained after a complex process described in [11]. The operation in which the encrypted message $c(x) \in R/q$ is obtained is described by (8),

$$c(x) = e(x) + m'(x) \mod (q, x^n - 1) \tag{8}$$

In this encryption operation, the multiplication operation required to calculate $e(x)$ consumes the highest percentage of time in relation to the total encryption time [6,24].

For that reason, the goal of this work is to provide a new implementation, suitable for constrained devices, that accelerates the algorithm execution at the expense of a low increase in resources.

### 2.2. Hardware Implementation of Polynomial Multiplication

The multiplication described in (7) follows a cyclic convolution process that can be expressed by (9):

$$e_k = \sum_{i+j=k \bmod N} (h_j \cdot r_i) \mod q \tag{9}$$

where the polynomial degree is expressed as $N$; $e_k$ represents the $k$-th coefficient of $e(x)$; $h_j$ the $j$-th coefficient of $h(x)$; and $r_i$ the i-th coefficient of $r(x)$. It is trivial to prove that this operation requires $N \cdot N$ scalar multiplications to conclude. The first effort to accelerate this operation at the hardware level was reported in [25]. The main changes introduced in this work were (i) the replacement of the multiplication by adding coefficients of the public key, $h(x)$, to the temporal result for each nonzero element in the blinding polynomial, $r(x)$, now as a binary polynomial; and (ii) the substitution of $r(x)$ by $r_1(x) + r_2(x)$ reducing the total operation cycles to $(d_1 + d_2) \cdot N$, being $d_1$ and $d_2$ the nonzero coefficients of $r_1(x)$ and $r_2(x)$, respectively. This was a direct and basic implementation of the scalar multiplication that achieved good results at the hardware level. However, the techniques were progressively refined to achieve improved implementations, as in the case of [26], where the scalar multiplication was replaced by addition or subtraction depending on whether the coefficient $r_j$ was 1 or $-1$, respectively. Power reduction methods were used to provide a design specially adapted to security applications (RFIDs and sensor nodes) reducing the amount of resources and power needed in a physical implementation. On the other hand, due to the large number of zero elements contained in the polynomial $r(x)$, another important improvement in the hardware implementation of the polynomial multiplication was the exploitation of this fact in the works presented in [27,28]. They only consider the nonzero coefficients of $r(x)$ polynomial to implement the convolution product. The first one exploits their locations, whereas the second identifies the degrees of nonzero terms in the $r(x)$ polynomial during the load process. Using this consideration, the operation described in (9) can be expressed as shown in Algorithm 1. Therefore, the multiplication operation will require $(N \cdot nnz)$ scalar multiplications and $(N - nnz)$ clock cycles of null coefficients of $r(x)$, where $N$ is the degree of the polynomial and $nnz$ the number of nonzero elements (including 1 and $-1$) in the $r(x)$ polynomial. This enhancement significantly reduces the time required to complete the convolution operation.

---

**Algorithm 1** Accelerating the polynomial multiplication using nonzero elements

---

**for** $i = 0 : N - 1$ **do**
　　**if** $r_i \neq 0$ **then**
　　　　**for** $k = 0 : N - 1$ **do**
　　　　　　$j = \mathrm{mod}(k - i, N)$
　　　　　　$e_k = e_k + (h_j \cdot r_i)$
　　　　**end for**
　　**end if**
**end for**

---

A parallelization process of this convolution product can be carried out by adding different scalar multiplications per operation cycle, considerably reducing the overall operation time. With the consequent cost in terms of resources, the work in [29] presents a fully parallelized hardware in which $N$ scalar multiplications are performed in each cycle using a linear feedback shift register (LFSR) structure. In [17,30], an improvement over [29] is proposed, where the multiplication operation is accelerated by analyzing when two, three, or four consecutive zeros are presented in the obfuscation polynomial $r(x)$. In [17,31], it was analyzed whether the total number of cycles, or in other words, the total

time required to complete the multiplication operation when it is accelerated considering consecutive elements can generate a security breach by using timing attacks that could allow the coefficients of the obfuscation polynomial $r(x)$ to be induced. On the other hand, the use of fully parallel structures to reduce the number of cycles implies a high cost in terms of resources, which constrained implementations cannot afford. To solve these drawbacks, a low-resource architecture for NTRUEncrypt based on a partial parallelization of the scalar multiplications that does not present any security breach against timing attacks is proposed in [18]. In this case, the operation could be accelerated using only the $2 \cdot dr$ nonzero coefficients of the polynomial $r(x)$ ($dr$ is the number of coefficients that are 1 and $-1$), i.e., $r(x) \in \mathcal{T}(dr)$, resulting in a number of operations equal to $(N \cdot 2dr) + (N - 2dr)$. More recently, the work presented in [32] described a full hardware implementation of the three multiplications required in NTRU Round 3. However, such implementation is not performed on the software structure of the cryptosystem presented in [11], using a large amount of resources to complete the operation in parallel, making its implementation on resource-constrained devices unfeasible.

## 3. Robust Acceleration against Timing Attacks

In the version of NTRU submitted to the third round of the NIST PQC contest, as described above, $r(x) \in \mathcal{T}$, i.e., the number of nonzero coefficients, is not fixed, making it impossible to predict the number of ones, minus-ones, and zeros before generating the polynomial. Using a pseudocode such as the one described in Algorithm 1 would cause a security breach if timing attacks were performed (by disclosing the information of the number of nonzero coefficients contained in $r(x)$). The goal of this proposal is to achieve some degree of acceleration of the $N \cdot N$ clock cycles of the convolution product without revealing any sensitive information of the polynomial $r(x)$.

Since it is not possible to know in advance the number of nonzero elements that the polynomial $r(x)$ will have, the strategy to accelerate this process consists of the evaluation of a significant number of $r(x)$ possible generations in order to establish an upper limit. For the set of parameters *ntruhps2048509*, Figure 1 shows a distribution of the number of nonzero elements when the polynomial $r(x)$ is generated $10^6$ times. The red line represents the maximum number of nonzero elements obtained. In this case, only one case reaches the value 390 (a probability of 0.0001%). In other words, establishing this threshold as a limit to perform the multiplication operation would not generate any security breaches since there is no temporal distinction between different generations of $r(x)$. The hardware multiplier, therefore, will have to operate for at least that number of cycles so as not to raise suspicions about the number of nonzero elements contained in the polynomial $r(x)$.

In a real implementation, for a specific use case, margins must be established from this threshold of nonzero coefficients. In this particular case, for the specific set of parameters the confidence margin is set by the blue line in Figure 1, with a value of 400 (selected by the designer). Fitting the number of nonzero coefficients to a normal distribution, this design threshold corresponds to a probability of 0.000034% (approximately 1 in 3 million). Thus, we can consider that it is practically impossible to obtain a polynomial $r(x)$ with more nonzero coefficients than the 400 established. In the case that this happens, establishing some design mechanism that allows for regeneration of the polynomial $r(x)$ at the software level would solve this remote problem. Hereinafter, this design threshold value will be referred to as $max_{coef}$. This limit of coefficients will include a large number of nonzero coefficients and a small number of zero coefficients. The procedure of threshold estimation should be repeated for the rest of the NTRU parameter set described in [11]: *ntruhps2048677*, *ntruhps4096821*, and *ntruhrss701*.

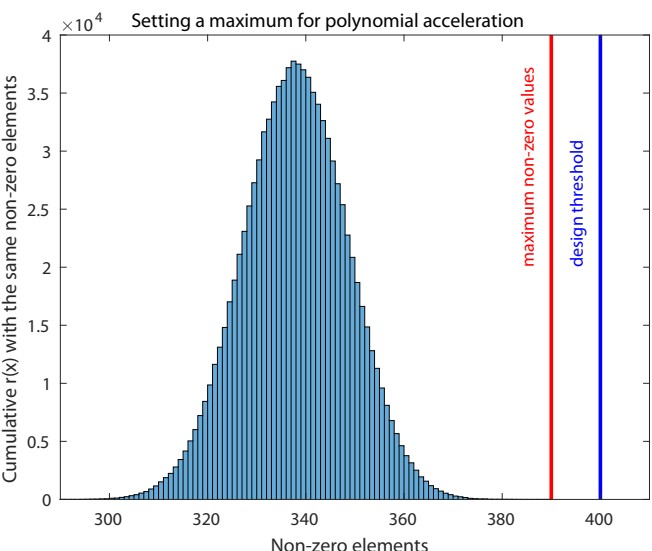

**Figure 1.** Distribution of nonzero elements in different $r(x)$ generations. The red line represents the maximum obtained, while the blue line defines a confidence threshold for the implementation.

These changes can be applied to the operation of the multiplier, so the pseudocode presented in Algorithm 1 can be upgraded to the pseudocode of Algorithm 2. In this case, the total number of cycles $total_{cycles}$ required to complete the operation will be $total_{cycles} = N * max_{coef} + (N - max_{coef})$. Setting a fixed number of clock cycles avoids the possible leakage of timing information that would allow the number of nonzero coefficients of $r(x)$ used in the multiplication to be known. The inclusion of this countermeasure does not entail a significant reduction in acceleration that can be achieved by taking advantage of the fact that $r(x)$ is ternary. To complete the operation, it is necessary to know the number of nonzero coefficients of the polynomial $r(x)$, $nnz$. This number must be calculated for each polynomial in each execution, which can be carried out and stored internally at a stage prior to the multiplication phase. Therefore, with this information, the number of zeros to be computed as if they were nonzero elements, i.e., those below the threshold $max_{coef}$, corresponds to the variable defined as $number_{zeros-max}$ or $nzm$. Therefore, in order to avoid any leakage of timing information, the acceleration produced by the elimination of the cycles corresponding to the null coefficients of $r(x)$ will only take place when the threshold determined by $max_{coef}$ is exceeded. Thus, the pseudocode is designed so that the operation is only performed if one of the following conditions occurs: (1) the maximum number of zeros $nzm$ has not yet been reached; and (2) the limit has been exceeded and the coefficient of $r(x)$ is not zero.

---

**Algorithm 2** Accelerating the polynomial multiplication considering nonzero elements and avoiding timing attacks

---

$nzm = max_{coef} - nnz$

**for** $i = 0 : N - 1$ **do**
    **if** $r_i = 0$ **then**
        $nz = nz + 1$
    **end if**
    **if** $nz \leq nzm$ **or** ($r_i \neq 0$ **and** $nz > nzm$) **then**
        **for** $k = 0 : N - 1$ **do**
            $j = \text{mod}(k - i, N)$
            $e_k = e_k + (h_j \cdot r_i)$
        **end for**
    **end if**
**end for**

---

## 4. IP Module Design and Integration

### 4.1. Design of the Arithmetic Unit

The design goal of this hardware-level multiplier is to be fully compatible with the reference version of NTRU submitted to the third round of the NIST PQC contest, so that software routines can be interchanged with hardware routines in a much easier and more efficient way. This implies that some particularities in both arithmetic and data types at the software level must be taken into account for software/hardware implementation. In the NTRU Round 3 scheme [11], the coefficients of the polynomials $r(x)$ and $h(x)$ are computed as modulus 2 and modulus 2048, respectively. The efficient arithmetic unit (AU) presented in [16–18] has been slightly modified. The new architecture uses a logic gate AND instead of a multiplexer. The AU is controlled by the $r_i$ coefficient, whose operation is summarized in Table 1 and shown in Figure 2.

**Table 1.** AU operation in function of $r_i$.

| $r_i$ | Operation |
|---|---|
| 00 | $e_{out_k} = e_{in_k}$ |
| 01 | $e_{out_k} = e_{in_k} + h_j$ |
| 11 | $e_{out_k} = e_{in_k} - h_j$ |

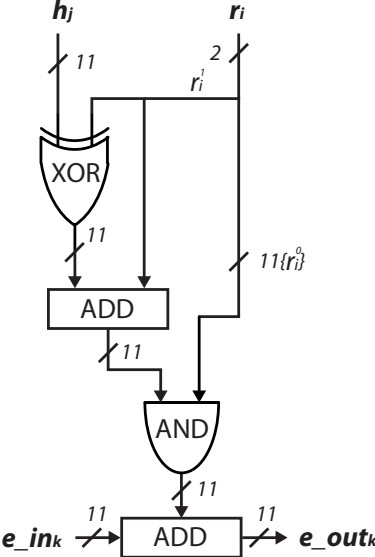

**Figure 2.** Block diagram of the arithmetic unit designed in this work.

### 4.2. Core Design

The architecture of the polynomial multiplier used in the NTRU Round 3 encryption scheme is described in this section. Although this work is focused on the use of encryption for the *ntruhps2048509* parameter set described in [11], the IP module is configurable for any other parameter set. The module was developed using the RTL-based design flow provided by Xilinx Vivado tools, and the Verilog Hardware Description Language (HDL) was used for hardware description. A simplified block diagram that contains the main functional blocks necessary for the hardware implementation of the polynomial multiplication operation is shown in Figure 3.

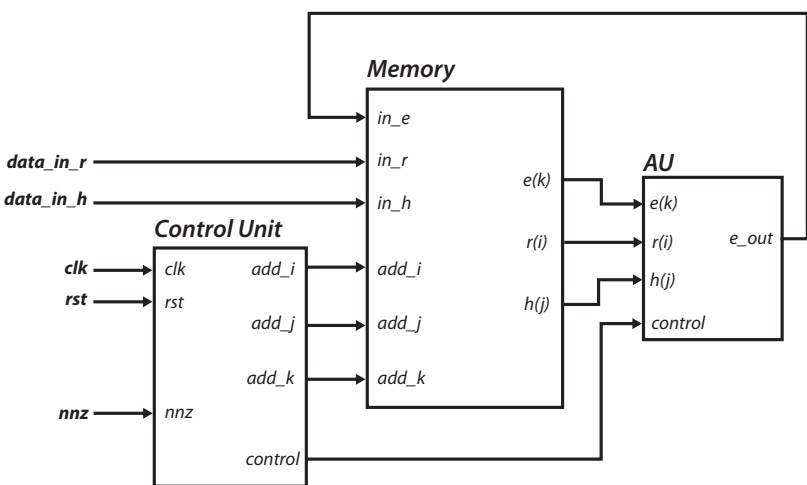

**Figure 3.** Simplified block diagram of the hardware polynomial multiplier architecture.

Multiplier operation is coordinated by the Control Unit block. It manages all operations in different phases: *Load* coefficients, *Operate*, and *Read* result. It generates the indices $i$, $j$, and $k$, which are used in each phase as memory addresses in the Memory block. This component contains the coefficients of the input polynomials, $r(x)$ and $h(x)$, stored in the *Load* phase. It also stores the partial results during the *Operate* phase, as well as the resulting polynomial, $e(x)$, that will be provided by the module in the *Read* phase. The memories included in this block were implemented as dual-port memories using the block RAMs (BRAMs) usually available in many programmable devices. During the *Operate* phase, the Control Unit obeys the operation described by the pseudocode in Algorithm 2; while the AU carries out either addition or subtraction over $h_j$, depending on the value of $r_i$, updating the partial results $e_k$. The number of total clock cycles required to complete the operation, $CC_{op}$, including the clock cycles for coefficient loading, $CC_{load}$, and reading, $CC_{read}$ (both require one clock cycle per coefficient), and the number of total clock cycles for the multiplication, $CC_{mult}$, is described by (10). The number of coefficients, $N$, and the threshold fixed to avoid timing attacks, $max_{coef}$, are the main factors determining the overall system runtime.

$$
\begin{aligned}
CC_{op} &= & CC_{load} + CC_{mult} + CC_{read} \\
&= & 2N + N \cdot max_{coef} + (N - max_{coef}) \\
&= & N \cdot (max_{coef} + 3) - max_{coef} \\
&\approx & N \cdot max_{coef}
\end{aligned}
\tag{10}
$$

*4.3. Parallelizing the Multiplication Process*

In the scheme described above, the number of clock cycles required for the system to complete the multiplication operation can be further reduced. A possible solution in this proposed scheme is acceleration using AUs operating in parallel. The strategy is the inclusion of $M$ AUs, where $M$ is the parallelization degree of the system so that the multiplication module can operate on $M$ coefficients per cycle. To introduce the ability to parallelize the system through the $M$ parameter, the pseudocode presented in Algorithm 2 is redefined in Algorithm 3, in which the the total clock cycles due to the operation of the AUs is reduced by a factor $M$. This reduction will depend on the degree of the polynomial and the number of AUs instantiated in parallel. Therefore, the number of total clock cycles of the parallel operation, $CC_{op}^{*}$, including both the loading and reading of the coefficients as well as the multiplication operation, $CC_{mult}^{*}$, is reduced according to (11). In this case, the estimation of the total number of total clock cycles is directly related to the $M$ parameter. Large values of $M$ involve a considerable reduction in the number of clock cycles used for

coefficient multiplication. Thus, the times required for writing and reading the coefficients to and from the module should not be underestimated when evaluating the total time for the module operation.

---

**Algorithm 3** Parallelizing the polynomial multiplication considering nonzero elements and avoiding timing attacks

---

$M \leftarrow$ Parallelization parameter
$nzm = max_{coef} - number_{non-zero}$

**for** $i = 0 : N - 1$ **do**
   **if** $r_i = 0$ **then**
      $nz = nz + 1$
   **end if**
   **if** $nz \leq nzm$ **or** $(r_i \neq 0$ **and** $nz > nzm)$ **then**
      **for** $k = 0 : M : N - 1$ **do**
         $j1 = \mod(k - i, N)$
         $e_k = e_k + (h_{j1} \cdot r_i)$
         $j2 = \mod(k - i + 1, N)$
         $e_{k+1} = e_{k+1} + (h_{j2} \cdot r_i)$
         $\vdots$
         $jM = \mod(k - i + M - 1, N)$
         $e_{k+M-1} = e_{k+M-1} + (h_{j1} \cdot r_i)$
      **end for**
   **end if**
**end for**

---

$$
\begin{aligned}
CC_{op}^* &= & CC_{load} + CC_{mult}^* + CC_{read} \\
&= & 2N + \left\lceil \frac{N}{M} \right\rceil \cdot max_{coef} + (N - max_{coef}) \\
&= & 3N + max_{coef} \cdot \left( \left\lceil \frac{N}{M} \right\rceil - 1 \right)
\end{aligned}
\tag{11}
$$

In Figure 4, a theoretical calculation to compare the strategy with $max_{coef} = N$ and with $max_{coef} = 400$ of total clock cycles required to perform the multiplication operation in the function of $M$ is shown. For its representation, a continuous function of the number of clock cycles as a function of $M$ has been taken into account. Essentially, it is the representation of the expression presented in (11). It can be seen how the number of cycles decays very fast until about $M = 10$, where the clock cycles needed for loading and reading the coefficients begin to be significant. Furthermore, the yellow line shows the difference of clock cycles between the two strategies of $max_{coef}$ in terms of percentage. The method detailed in Section 3 allows for a reduction between 17% and 20% of the operation time in parallelized solutions.

Since the acceleration strategy is based on the parallelization of the operation, at the hardware level, it is necessary to replicate the AUs $M$ times. This also involves a replication of the Memory block, which increases the size in order to provide both the addresses and coefficients to complete the multiplication correctly. The implementation strategy followed for the AUs, bus sizes, and Memory block replication was detailed in [18]. The Control Unit must be also modified in order to generate the index addresses, taking into account that there is more than one coefficient operating in the same clock cycle. The block diagram of the hardware implementation considering acceleration is shown in Figure 5.

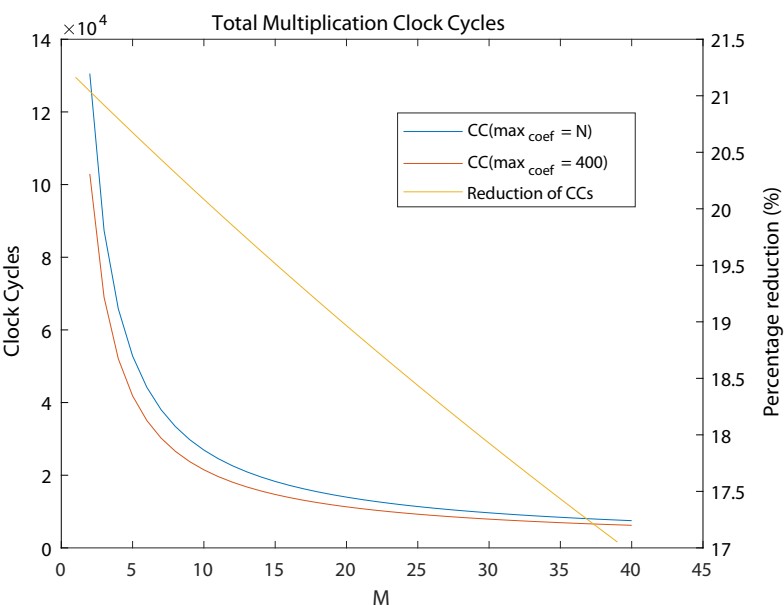

**Figure 4.** Comparison in terms of clock cycles between strategies with $max_{coef} = N$ and $max_{coef} = 400$ versus $M$.

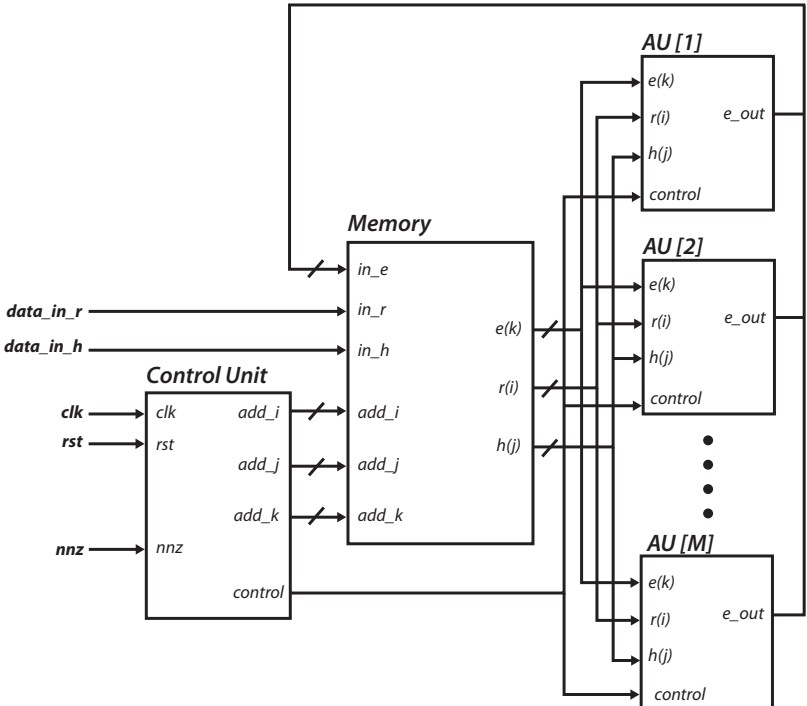

**Figure 5.** Block diagram of the hardware polynomial multiplier architecture considering parallelization.

### 4.4. Embedded System Integration

The hardware architecture detailed above must be interconnected with a processing system (PS) to build a hybrid implementation. The IP module and the PS are connected using standard interconnection buses that facilitate design reusability. In this case, the cryptosystem defined at the software level and executed on a general-purpose processor sends the coefficients of the input polynomials; thus, this communication protocol also has to be designed following these considerations. In order to use the proposed multiplier on system-on-chip (SoC) solutions—for example, those that incorporate an ARM processor—the most suitable option is to use the Advanced eXtensible Interface (AXI) bus.

In this work, the AXI interconnection interface is implemented through an AXI4-Stream interface. For a correct synchronization between the data sent by the processor and the IP module, it is necessary to instantiate first-in, first-out (FIFO) structures both at the input and output. Figure 6 shows the final multiplication module, in which the FIFOs have been integrated into the blocks called *Data In* and *Data Out*. The memory addresses used to store the $r(x)$ and $h(x)$ coefficients in the *Load* phase and to read the $e(x)$ coefficients in the *Read* phase are internally provided by the Control Unit.

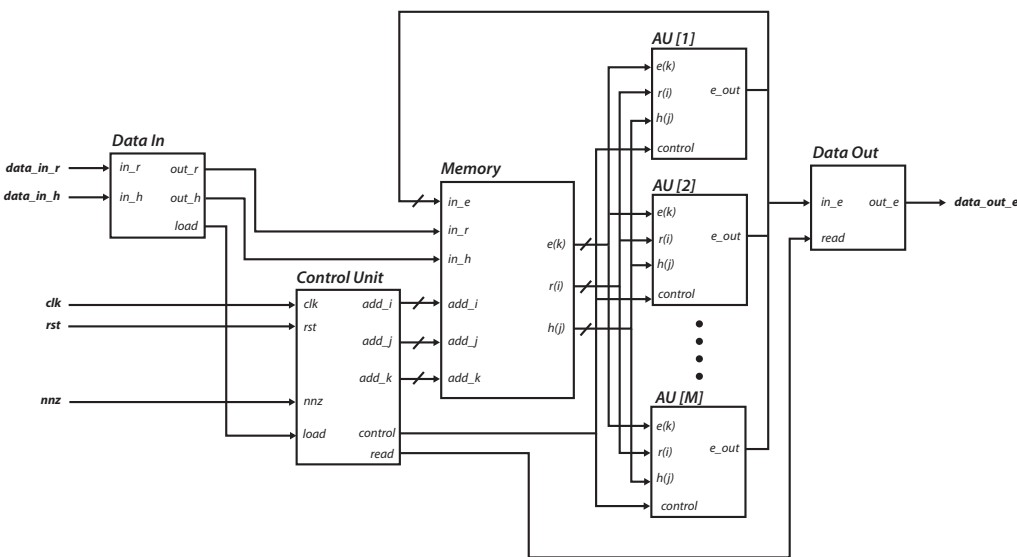

**Figure 6.** Block diagram of the hardware polynomial multiplier architecture, considering parallelization and including AXI4-Stream interconnection interfaces.

The use of standard communication protocols such as AXI4-Stream makes the module fully integrable with other devices. Apart from that, the design of the IP module is completely reusable, being able to change implementation parameters related to the cryptosystem, such as the degree of the polynomial, $N$, as well as others that influence the timing performance, such as $max_{coef}$ or $M$. Therefore, first, it is fully functional on any of the parameter sets defined in the NTRU Round 3 version, and second, the implementation can be adapted to be more or less restrictive in terms of area and timing performance depending on the constraints imposed on the system to be implemented.

Specifically, this work has used the development board PYNQ-Z2, which integrates the Xilinx Zynq-7000 SoC (XC7Z020-1CG400C). The device consists of a PS that operates at 650 MHz and includes a dual-core ARM-Cortex-A9, along with programmable logic (PL) from the Xilinx Artix-7 FPGA family. The selected development board supports the Python Productivity for Zynq (PYNQ) environment [33]. This describes a framework for Python, which operates on an embedded Linux operating system. The framework streamlines the process of connecting hardware modules with software components.

At the hardware level, to complete the interconnection scheme between the PS and the IP module, shown in Figure 7, some extra modules are required: the Direct Memory Access (DMA), AXI Interconnect, and AXI SmartConnect blocks handle the exchange of information related to the polynomial coefficients. The IP module receives the coefficients of the input operands at the beginning of the operation and sends the coefficients of the result at the end of the operation from/to the DMA blocks, respectively. For the design and implementation of the IP module, the Vivado 2020.2 design tool was used. This facilitates both the design and its integration in embedded systems, making the design of a parametrizable IP module possible.

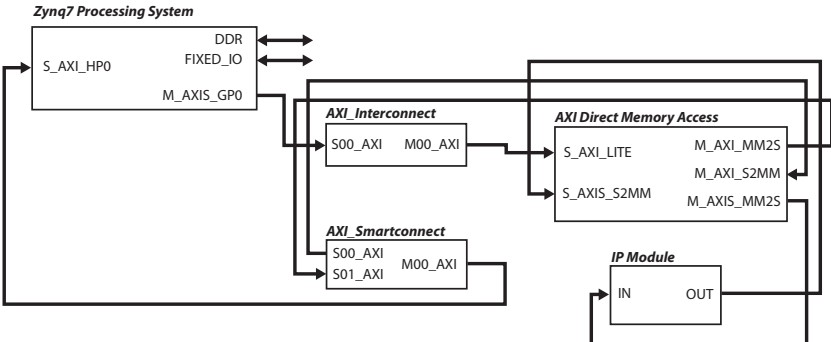

**Figure 7.** Block diagram of the complete embedded system and the necessary blocks to interconnect the IP module with the Zynq Processor.

At the software level, the NTRU Round 3 version implementation used was the optimized one presented in [11]. This is a complete C implementation of the NTRU public key scheme presented in the third round of the NIST contest. For this work, a completely new set of tests was designed, directly providing results relative to the acceleration of the encryption scheme using dedicated hardware. The implementation of NTRU in C was adjusted for the PYNQ environment by utilizing C-API, which is made available in [34]. This C-API offers a comprehensive set of C routines that can be compiled to produce executable code. PYNQ C-API includes features that make it easier to load bitstreams and communicate with hardware blocks located on the PL of the Zynq device through memory-mapped and shared memory mechanisms. The utilization of these features not only simplifies the creation of software drivers required to manage the hardware multipliers, but also eases the programming of a series of tests to validate and characterize their operation.

## 5. Results

In this Section, the multiplication module is evaluated in terms of resource consumption, and the acceleration factors in both multiplication and encryption processes are analyzed. Different versions of the multiplication module for NTRU Round 3 were implemented in this work for multiple values of $M$ and considering or not the limit established by $max_{coef}$.

### 5.1. Resource Consumption

Using the parametrizable IP Module, it is possible to modify the degree of the polynomial, $N$, and the number of AUs in parallel, $M$, as well as the maximum number of coefficients for acceleration, $max_{coef}$. For the evaluation, the parameter set *ntruhps2048509* was used. In other words, some parameters that characterize the IP module are set, e.g., the degree of the polynomial, $N = 509$. The IP module was implemented, varying the values of $max_{coef}$ and $M$ for this specific parameter set. Although it is possible to select any value, in order to comprehensively cover the entire possible range of the parallelization coefficient, $M$, that generates the different parallel implementations, all powers of two starting from 2 up to 256, both inclusive, were used.

Table 2 shows the use of lookup tables (LUTs), flip-flops (FFs), and block RAMs (BRAMs) of the implementation with $max_{coef} = 400$. For the case of $max_{coef} = N$ there is an increase in execution time of around 10%, while the resource occupation remains the same, which is the first remarkable result. For the case of $max_{coef} = 400$, it was expected that it would require more logic to implement all the changes related to the acceleration of multiplication in the control module. However, this increase is minimal. In terms of the parallelization coefficient, $M$, Table 2 shows that as the value doubles, the resource consumption close to doubles in the case of LUTs and BRAMs, but not in the case of FFs, where the trend is simply upward. For the highest value of $M$, 256, while there is hardly any variation in occupancy relative to FFs, around 30% of LUTs and 90% of BRAMs are

occupied. With the increase in $M$, the resources required for the implementation also increase, being even more critical to resource consumption. Summarizing the results, up to an index $M = 32$, occupancy is approximately below 10%, being even around 0.5–2% in total for $M = 4$.

**Table 2.** Comparison of the IP module resource occupation and timing performance for the implementation with $max_{coef} = 400$.

| | | | $max_{coef} = \mathbf{400}$ | | | |
|---|---|---|---|---|---|---|
| **M** | **LUTs** | **FFs** | **BRAM** | **Clk (MHz)** | $CC_{mult}$ | **Latency (μs)** |
| 1 | 166 | 96 | 1.5 | 96.53 | 203,709 | 2110.43 |
| 2 | 241 | 121 | 2.5 | 97.28 | 102,109 | 1049.68 |
| 4 | 344 | 119 | 4.5 | 98.04 | 51,309 | 523.35 |
| 8 | 658 | 205 | 4.5 | 94.43 | 25,709 | 272.25 |
| 16 | 1032 | 291 | 8.5 | 94.43 | 12,909 | 136.70 |
| 32 | 1939 | 471 | 16.5 | 92.94 | 6509 | 70.03 |
| 64 | 4452 | 826 | 32.5 | 86.43 | 3309 | 38.28 |
| 128 | 8668 | 1545 | 64.5 | 79.93 | 1709 | 21.38 |
| 256 | 17,505 | 3017 | 128.5 | 77.77 | 909 | 11.70 |

With respect to the embedded system integration, there is a minimum use of resources necessary to instantiate the DMAs and the communication infrastructure, adding a resource consumption of approximately 3750 LUTs, 5300 FFs, and 2 BRAMs. The percentage of occupation due to all these connections in the embedded system is independent of the size of the IP module and clearly notable for the small values of $M$.

Table 3 firstly shows the comparison in terms of area occupation and timing performance, with respect to the work presented in [32], which is based, as is this work, on the NTRU submitted to NIST PQC Round 3. The comparison between the two implementations shows that the one presented in this work has less resource occupation and a better timing performance. Moreover, this work provides both a detailed analysis of the potential risks of timing attacks associated with this NTRU version and a balance between cost and performance, which is crucial in the context of IoT. Thus, our contribution in this regard is a strong and reliable version for IoT that requires minimal additional resources. In fact, 256 AUs were used for the comparison. However, this number can be reduced to any value that satisfies the occupancy constraints of a particular IoT device. Additionally, Table 3 includes several works in the literature, such as [17,18,30,31], which present implementations for the NTRU version standardized in IEEE-1363.1, different from the one used in this work. The works presented in [17,30,31] implement parallel structures to perform the NTRU multiplication, while the work presented in [18] shows a serial implementation that is very close to our work.

**Table 3.** Resources and timing performance comparison between this work, a recent work of the latest NTRU version, and other works of the previous standard.

| **NTRU Version** | **Work** | **LUT** | **FF** | **#CC** | **Latency (μs)** | **#AU** |
|---|---|---|---|---|---|---|
| Round-3 | Our Work | 17,505 | 3017 | 1018 | 11.70 | 256 |
| | [32] | 56,218 | 21,406 | 821 | 12.32 | 509 |
| IEEE-1363.1 | [17] | 29,194 | 19,096 | 245 | 3.23 | 541 |
| | [18] | 603 | 90 | 7107 | 71.07 | 8 |
| | [30] | 30,300 | - | 343 | 3.62 | 541 |
| | [31] | 38,240 | - | 541 | - | 541 |

### 5.2. Analysis of Acceleration Factors

Another aspect that requires detailed analysis is the time reduction that the IP module achieves in both the multiplication process and the encryption of the NTRU Round 3 software version. That is, how much it allows the operation of the multiplication to speed up when comparing a full software version, executed on the embedded system processor, versus the use of the hardware IP module. Table 4 shows the acceleration produced by the use of the IP module with both strategies of $max_{coef}$ for both the multiplication and the complete encryption process, respectively. For each value of $M$, 1000 tests were performed, with software and hardware mean times obtained. The values of both the multiplication and the encryption time in the software are the mean values of the whole tests performed (18,000 in total).

Using this IP module in the NTRU Round 3 version with $max_{coef} = 400$ and with $max_{coef} = N$, accelerations of over six and five times, respectively, in both multiplication and encryption, are achieved for $M = 1$. As the value of $M$ increases, the hardware operation time is reduced, also increasing the acceleration both in the multiplication operation and in the encryption scheme. For the multiplication case in Table 4, a maximum acceleration factor of 72 times is achieved with the use of the maximum degree of parallelization. The results also reveal that the differences between both strategies of $max_{coef}$ are more significant for intermediate $M$ values. The strategy of $max_{coef} = 400$ presents certain temporal advantages, reducing around 10% of the time required with respect to the second one, $max_{coef} = N$. As is also observed, a limit in the acceleration at around 65–70 times is reached when $M$ is increased. This is mainly due to the fact that the operating time accelerates to such an extent that the times required to exchange coefficients to and from the IP Module are no longer insignificant. This behavior was already predicted earlier in Figure 4.

**Table 4.** Multiplication and encryption acceleration using the hardware implementation with $max_{coef} = 400$ and $max_{coef} = N$, with respect to the time required for the software.

| | | $max_{coef} = 400$ | | $max_{coef} = N$ | | | $max_{coef} = 400$ | | $max_{coef} = N$ | |
|---|---|---|---|---|---|---|---|---|---|---|
| **M** | **SW (µs)** | **HW (µs)** | **Acc. (x)** | **HW (µs)** | **Acc. (x)** | **SW (µs)** | **HW (µs)** | **Acc. (x)** | **HW (µs)** | **Acc. (x)** |
| 1 | | 2219 | 6.47 | 2772 | 5.18 | | 2347 | 6.16 | 2900 | 4.95 |
| 2 | | 1205 | 11.91 | 1486 | 9.66 | | 1334 | 10.76 | 1614 | 8.89 |
| 4 | | 699 | 20.53 | 842 | 17.05 | | 826 | 17.38 | 972 | 14.77 |
| 8 | 14,354 | 445 | 32.26 | 509 | 28.20 | 14,468 | 575 | 24.96 | 639 | 22.46 |
| 16 | | 316 | 45.42 | 350 | 41.01 | | 444 | 32.33 | 479 | 29.97 |
| 32 | | 254 | 56.51 | 271 | 52.97 | | 384 | 37.38 | 400 | 35.88 |
| 64 | | 224 | 64.08 | 231 | 62.14 | | 352 | 40.78 | 360 | 39.87 |
| 128 | | 207 | 69.34 | 209 | 68.68 | | 335 | 42.85 | 338 | 42.47 |
| 256 | | 198 | 72.49 | 199 | 72.13 | | 327 | 43.90 | 328 | 43.76 |
| | | **Multiplication** | | | | | **Encryption** | | | |

In the case of the encryption scheme in Table 4, since the software time required for multiplication (14,354 µs) is equivalent to about 95% of the software time required for encryption (14,468 µs), the time reductions follow the same trend. This can be verified where as in the case of multiplication—the central values of $M$ are where the difference between the use of both values of $max_{coef}$ is the greatest. On the other hand, as the degree of parallelization increases, a limit as in the case of multiplication is reached. In this case, the acceleration factor obtained is close to 43. The difference with respect to multiplication is mainly due to the fact that in the encryption process of the NTRU Round 3 version, other software functions different from the polynomial multiplication are executed, which are necessary and should not be disregarded. No matter how much the multiplication function is accelerated, the execution of these operations requires time that cannot be avoided.

This can be seen in Figure 8, which shows the comparison in terms of acceleration that occurs for the $max_{coef} = 400$ strategy in both multiplication (orange line) and full encryption (blue line). The first values of the parallelization coefficient, $M$, make the speed-up grow very rapidly, while from $M = 32$ onward, the speed-up seems to stabilize. As the value of $M$ increases, the separation between the accelerations produced by the multiplication and the encryption process also increases, mainly because the times invested in the multiplication process start to become less significant, while all encryption operations different from the multiplication start to become more significant in time.

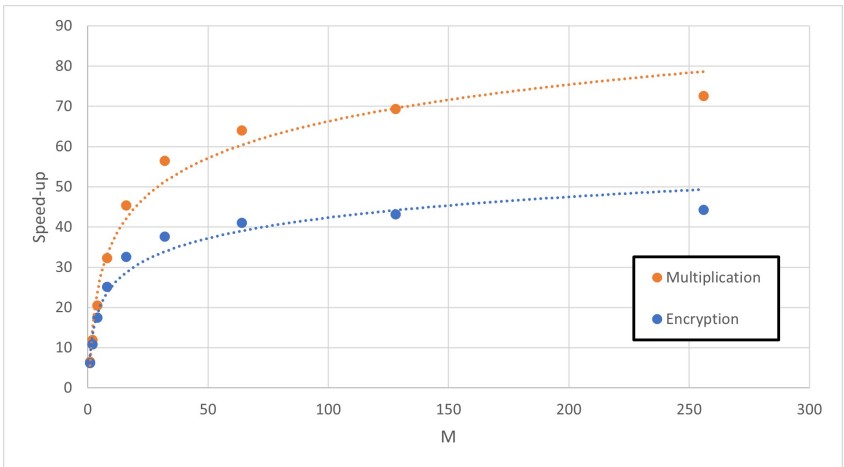

**Figure 8.** Illustrative example of the speed-up factor versus $M$ for multiplication and encryption processes.

*5.3. Optimizing Area and Acceleration*

In constrained devices, it is especially important to analyze the resources required to implement a function. In other words, evaluating how the specific use of each resource is capable of separately speeding up the algorithm to a greater or lesser extent is fundamental for establishing certain design decisions regarding area optimization. Since in this work we previously presented results related to LUT, FF, and BRAM occupancy, as well as accelerations as a function of the parallelization index $M$, it is possible to study how this parameter $M$ jointly affects the resources used and acceleration obtained. For this purpose, a new figure of merit, *Efficiency*, $E$, is defined for each type of resource as the quotient between the acceleration obtained and the resources (LUTs, FFs, and BRAMs) used for each $M$:

$$E_{LUT}(M) = \frac{Acc.(M)}{LUT(M)} \tag{12}$$

$$E_{FF}(M) = \frac{Acc.(M)}{FF(M)} \tag{13}$$

$$E_{BRAM}(M) = \frac{Acc.(M)}{BRAM(M)} \tag{14}$$

Optimization of this figure of merit occurs when few resources are used and high acceleration is obtained. This combination causes the efficiency to tend towards higher values. For simplicity in evaluating these results, only the resources used to implement the IP module are used. An analogy could easily be drawn in terms of occupancy with the embedded system. In the same way, the temporal results related to the acceleration of the multiplication are used. Again, an analogy could be established with encryption, but the conclusions to be drawn would be completely identical. Thus, using the results presented in Table 2 concerning the number of resources used to implement the IP module, as well as the data in Table 4 concerning the acceleration of multiplication, both considering the strategy of $max_{coef} = 400$ and $max_{coef} = N$, it is possible to quantify the efficiency parameter defined above.

Results are shown in Figure 9a,b for both strategies of $max_{coef}$. For simplicity and ease of visualization, only the data up to $M = 32$ are shown. Results in blue are related to efficiency in terms of LUTs; in orange, FFs; and in green, BRAMs. In general, in both cases, it can be observed that both LUTs and FFs have a very similar trend, with BRAMs being the elements that seem to follow a different trend. The maximum LUT optimization occurs at around $M = 4$ for both cases. That is, for that value of $M$, the resource expenditure associated with LUTs is the one that provides the highest accelerations with the lowest occupancy. Following the same analogy, for FFs, it occurs around at the same $M = 4$ for both cases, whereas for BRAMs, it occurs around at $M = 8$ for both cases. Thus, for example, for $M = 4$, we have a considerable increase in efficiency for both LUTs and FFs, but not so for BRAMs. The opposite case occurs at $M = 8$, where this time, the optimized resource is the BRAMs, while LUTs and FFs lose efficiency.

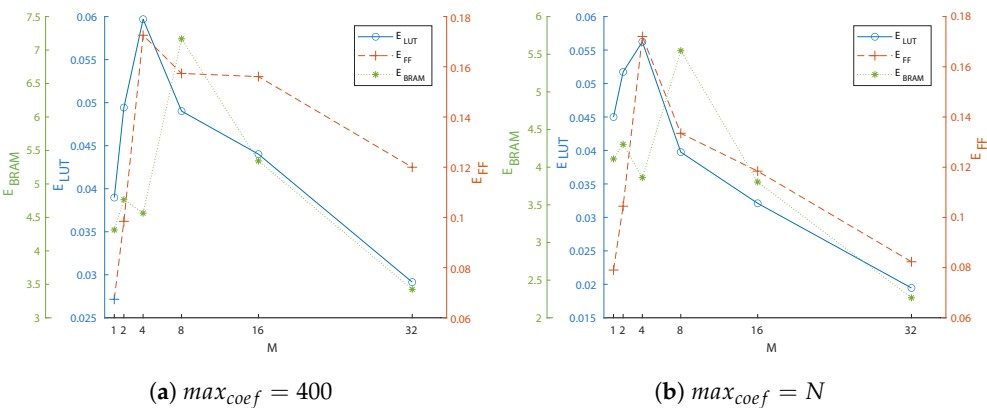

(**a**) $max_{coef} = 400$        (**b**) $max_{coef} = N$

**Figure 9.** Resource efficiency versus $M$ ($M$ being power of 2) in the multiplication process.

In order to complete the study of the most efficient implementation, a new set of IP modules was implemented for all values of $M$ between 3 and 16 for both strategies of $max_{coef}$. Table 5 shows the results relative to the IP module resource use for the strategy of $max_{coef} = 400$. As was mentioned before, the strategy of $max_{coef} = N$ increases the execution time by around 10%, keeping the resource occupation at the same level. In terms of timing performance, Table 6 shows the results of the multiplication acceleration using these new implementations of the IP module. Thus, the Efficiency can be recalculated and completed using these new results. Figure 10a,b shows the Efficiency for LUTs, FFs, and BRAMs in the strategies $max_{coef} = 400$ and $max_{coef} = N$, respectively, for $M$ values between 2 and 16, both inclusive.

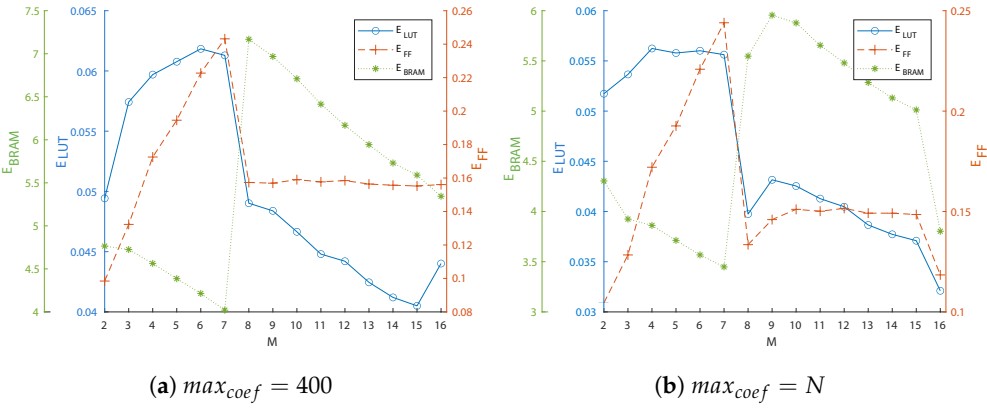

(**a**) $max_{coef} = 400$        (**b**) $max_{coef} = N$

**Figure 10.** Resource efficiency versus $M$ ($M$ between 4 and 16) in the multiplication process.

**Table 5.** Comparison of the IP module resource occupation and timing performance for the implementation with $max_{coef} = 400$ for the extended results.

| | | | $max_{coef} = 400$ | | | |
|---|---|---|---|---|---|---|
| **M** | **LUTs** | **FFs** | **BRAM** | **Clk (MHz)** | $CC_{mult}$ | **Latency (µs)** |
| 3 | 288 | 125 | 3.5 | 97.09 | 68,109 | 701.50 |
| 5 | 397 | 124 | 5.5 | 96.90 | 40,909 | 422.18 |
| 6 | 443 | 123 | 6.5 | 97.09 | 34,109 | 351.31 |
| 7 | 492 | 124 | 7.5 | 94.43 | 29,309 | 310.38 |
| 9 | 720 | 222 | 5 | 96.53 | 22,909 | 237.33 |
| 10 | 791 | 232 | 5.5 | 96.90 | 20,509 | 211.65 |
| 11 | 859 | 244 | 6 | 93.98 | 18,909 | 201.20 |
| 12 | 907 | 253 | 6.5 | 92.25 | 17,309 | 187.63 |
| 13 | 980 | 266 | 7 | 92.59 | 16,109 | 173.98 |
| 14 | 1043 | 276 | 7.5 | 90.66 | 14,909 | 164.46 |
| 15 | 1104 | 288 | 8 | 90.66 | 13,709 | 151.21 |

**Table 6.** Multiplication acceleration using the hardware implementation with $max_{coef} = 400$ and $max_{coef} = N$ with respect to the time required for the software for the extended results.

| | | $max_{coef} = 400$ | | $max_{coef} = N$ | |
|---|---|---|---|---|---|
| **M** | **SW (µs)** | **HW (µs)** | **Acc. (x)** | **HW (µs)** | **Acc. (x)** |
| 3 | | 868 | 16.49 | 1045 | 13.69 |
| 5 | | 595 | 24.12 | 703 | 20.42 |
| 6 | | 524 | 27.39 | 619 | 23.19 |
| 7 | | 476 | 30.16 | 555 | 25.86 |
| 9 | | 412 | 34.84 | 482 | 29.78 |
| 10 | 14,354 | 389 | 36.90 | 444 | 32.33 |
| 11 | | 373 | 38.48 | 423 | 33.93 |
| 12 | | 358 | 40.09 | 403 | 35.62 |
| 13 | | 345 | 41.61 | 388 | 36.99 |
| 14 | | 334 | 42.98 | 373 | 38.48 |
| 15 | | 321 | 44.72 | 358 | 40.09 |

**Table 7.** Summary of the *M* selected for the maximum efficiency in terms of resource occupancy and timing performance of the IP module for $max_{coef} = 400$ and $max_{coef} = N$.

| $max_{coef}$ | **Max. Eff.** | **M** | **LUTs** | **FFs** | **BRAM** | **Acc. (x)** |
|---|---|---|---|---|---|---|
| | **LUT** | 6 | 443 | 123 | 6.5 | 27.39 |
| **400** | **FF** | 7 | 492 | 124 | 7.5 | 30.16 |
| | **BRAM** | 9 | 658 | 205 | 4.5 | 32.26 |
| | **LUT** | 4 | 309 | 101 | 4.5 | 17.05 |
| **N** | **FF** | 7 | 465 | 106 | 7.5 | 25.86 |
| | **BRAM** | 9 | 690 | 204 | 5 | 29.78 |

Therefore, the values that provide the highest efficiency in the execution of the multiplication at the hardware level are for the strategy of $max_{coef} = 400$, $M = 6$ for LUTs, $M = 7$ for FFs, and $M = 8$ for BRAMs, as well as for the strategy of $max_{coef} = N$, $M = 4$ for LUTs, $M = 7$ for FFs, and $M = 9$ for BRAMs. Summarizing the results, Table 7 shows a selection of the data related to the IP module occupancy and the multiplication acceleration already presented in Table 2 and Table 4, respectively. From this selection of implementations, it is necessary to decide which is the most limited resource in a future cryptographic

implementation where this module fulfills a given function. It can be seen that LUTs are kept at around 1%, a significantly low occupancy value. In relation to FFs, the value of occupancy remains lower than 0.20%. Additionally, the high efficiency of the BRAMs has its origin in the fact that the number of BRAMs used does not follow a linear tendency, with fluctuations when $M$ is not a power of 2. In short, a certain trade-off between the timing performance of the system and the resources used is evident.

## 6. Conclusions

This work presents a hardware-dedicated architecture to accelerate—with very limited resource consumption—the polynomial multiplication in the encryption process of the NTRU version presented in the third round of the NIST contest. This polynomial multiplication is especially critical in the execution time of the encryption scheme. As with other implementations in the literature, the proposed solution contemplates a reduction in the number of clock cycles by considering the zero coefficients of the obfuscation polynomial $r(x)$. However, since the direct application of this procedure can introduce vulnerabilities in the security of the NTRU Round 3 version, a new method is proposed that allows the multiplication operation to speed up without compromising the security of the cryptosystem. The replication of similar AUs operating in parallel is also explored to further accelerate the process. This architecture is the basis of an IP module that includes standard interfaces based on the AXI4-Stream protocol communication, which facilitates hybrid HW/SW implementations on Xilinx's last-generation programmable devices.

Because of the reusability and integration facility of the IP module, a wide set of implementation results have been presented in order to compare the different alternatives in terms of resources and execution times. This eases integration in IoT environments that usually require strong area, power, and timing constraints. The study considers the implementation of the embedded systems in the Xilinx Zynq-7000 programmable device. The parameter set used for the comparison was the *ntruhps2048509* for the optimized version of the NTRU Round 3 submitted to the NIST PQC contest. Using the results obtained in terms of resources and timing performance, an efficiency analysis was carried out. The motivation of this evaluation was to obtain a strategy that requires less occupancy while reducing the execution time as much as possible. This is a critical aspect in applications with resource-constrained devices. In general, the results show that it is possible to accelerate the NTRU Round 3 encryption scheme between 30 and 45 times with respect to the reference software version, with a resource consumption of approximately 5%.

In conclusion, while this paper primarily discusses the application of the NTRU Round 3 version, the suggested approach and methodology can also prove to be valuable for the execution of other operations involving polynomial rings, such as the decryption process of NTRU Round 3, and the creation new cryptographic modules for algorithms chosen to be standardized by NIST.

**Author Contributions:** All authors contributed to conceptualization, investigation, and data curation-related tasks. Additionally, P.B. played an important role in supervision and funding acquisition. E.C.-R. proposed the methodology, implemented the prototypes, coded and executed the test programs, and coordinated the final manuscript edition, incorporating the suggestions of the rest of the authors. M.C.M.-R. worked out a first version of the manuscript. S.S.-S. contributed to the methodology, as well as to the design and validation of the hardware modules. All authors have read and agreed to the published version of the manuscript.

**Funding:** This research was supported in part by the SPIRS Project with Grant Agreement No. 952622 under the EU H2020 research and innovation programme and the ARES Project PID2020-116664RB-100 funded by MCIN/AEI/10.13039/501100011033 and the NextGenerationEU/PRTR. M.C.M.-R. holds a Postdoc fellowship from the Andalusia Government with support from PO FSE of EU. E.C.-R. is supported by the FPU20/03008 predoc grant from the Spanish government.

**Data Availability Statement:** Not applicable.

**Conflicts of Interest:** The authors declare no conflict of interest.

## Abbreviations

The following abbreviations are used in this manuscript:

| | |
|---|---|
| ANSI | American National Standards Institute |
| AU | Arithmetic Unit |
| AXI | Advanced eXtensible Interface |
| BRAM | Block Random Access Memory |
| CC | Clock Cycle |
| DH | Diffie-–Hellman |
| DMA | Direct Memory Access |
| DLP | Discrete Logarithm Problem |
| DSA | Digital Signature Algorithm |
| ECC | Elliptic Curve Cryptography |
| ENISA | European Union Agency for Cybersecurity |
| EU | European Union |
| FF | Flip-Flop |
| FIFO | First-In, First-Out |
| FPGA | Field-Programmable Gate Arrays |
| HDL | Hardware Description Language |
| HLS | High-Level Synthesis |
| HPS | Hoffstein, Pipher, and Silverman |
| HRSS | Hülsing, Rijnveld, Schanck, and Schwabe |
| HW | Hardware |
| IEEE | Institute of Electrical and Electronics Engineers |
| IP | Intellectual Property |
| IoT | Internet-of-Things |
| KEM | Key Encapsulation Mechanism |
| LUT | Look-Up Table |
| NIST | National Institute of Standards and Technology |
| NTRU | N-th-degree Truncated polynomial Ring Unit |
| PKC | Public Key Cryptography |
| PL | Programmable Logic |
| PQ | Post-Quantum |
| PQC | Post-Quantum Cryptography |
| PS | Processing System |
| PYNQ | PYthon Productivity for zyNQ |
| RAM | Random Access Memory |
| RSA | Rivest–Shamir–Adleman |
| RTL | Register-Transfer Level |
| SoC | System-on-Chip |
| SVP | Shortest Vector Problem |
| SW | Software |
| VLSI | Very Large-Scale Integration |

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
