# Peer review of "Timing-Attack-Resistant Acceleration of NTRU Round 3 Encryption on Resource-Constrained Embedded Systems"

_cryptography, doi:10.3390/cryptography7020029_

Round 1
Reviewer 1 Report
This study proposes a hardware-specific architecture to speed up the polynomial multiplication in the encryption process of the NTRU version presented in the third round of the NIST competition with very little resource consumption. My comments are as follows:
1. The authors should define every abbreviation in the paper. For example, NRRU in the abstract should be defined. Please check all the abbreviations in the paper.
2: The authors compare the performance of their work and other existing works in terms of some parameters in Table 4. It is right that LUTs and FF in their work are less than in [17] and [28]. However, as I see from Table 4, references [17] and [28] have less latency and #CC compared with the authors' method. It means the works in [17] and [28] are more suitable to be used in resource-constrained devices such as IoT. The increased number of FFs and LUTs is not that big of a problem because of the large and fast developments in electronics and DSP. I need authors to respond convincingly to this comment. They should show how their method outperforms the existing methods in IoT applications.
Reviewer 2 Report
Dear authors,
The work is interesting and is within the scope of the journal; verify that the Keywords appeared in the abstract description.
It defines IoT as a keyword, but it is not addressed within the document, it should probably be removed as a keyword.
High text match was found with this similar work
https://doi.org/10.3390/s22052057
care must be taken not to reuse text, even if it is from the same authors.
An adequate study of state of the art is found in the introduction.
The authors have published works on the NTRU cryptosystem; in this sense, it is necessary to define the main difference and improvement in the results discussion section against existing works.
The comparative tables are interesting, but I consider that they are excessive; it is necessary to verify which ones are relevant to demonstrate the contribution of the developed NTRU algorithm scheme.
Best Regards
Reviewer 3 Report
1. The techniques employed to maximize resource utilization while preventing temporal information leaks were not thoroughly examined by the paper's authors. Moreover, they omitted to explain the criteria by which the hardware multiplications were assessed or how they contrasted with alternative approaches. The long polynomial multiplication time that is the shortcoming in the NTRU cryptosystem is mentioned in the text, however, the methods utilized to speed up the process are not sufficiently described. The reduction of clock cycles that is suggested as a solution could result in security vulnerabilities, although this is not thoroughly addressed in the research. As a result, the paper lacks a thorough study of the strategies used to close the gap, and the proposed solution may not be reliable
2. The paper lists the contributions by specifying authors who worked on conceptualization, investigation, supervision, data-curation tasks, along with combined agreement on design and validation of modules.
3. The authors are trying to give a thorough overview of the subject and emphasize the key ideas in the introduction section. But it seems to be assumed that readers have some familiarity with the subject, which makes it challenging to understand for those who are not familiar with the topic. The introduction's background material is confusing and would benefit from more explanations of specific topics. The author touches on important issues, but some of the more complicated ideas lack clarity or explanation, which could make it difficult for readers to understand the general context of the article. To help the content be more understandable to a larger audience, the author could include. However, the related works have sufficient citations to back up the claims made. The shortcomings and restrictions of earlier research have been highlighted by the authors in a clear and succinct assessment of the literature that is currently available in the topic. The references included are current and pertinent, indicating that the writers did a comprehensive examination of the literature. Overall, the related works section is well-written and does a good job of laying the groundwork for the study's main research topic. more context and explanations.
4. A successful NTRU encryption and decryption strategy that can be used on embedded systems is the goal of the proposed study. The lattice-based public-key encryption system known as NTRU is regarded as a cornerstone of post-quantum cryptography (PQC). The scientific community has increased PQC to address this problem since in the near future the security of cryptographic protocols used in daily life will be undermined. The field of electronics engineering has an open issue in the design of hardware-efficient solutions. While implementing PQC-based algorithms, recent studies demonstrate the utilization of hybrid Hardware/Software (HW/SW) co-design approaches that combine flexibility and efficiency. The critical operation in the NTRU cryptosystem is the multiplication in the nth degree truncated polynomial ring, and thus, the proposed work focuses on its acceleration through hardware implementations.
5. The scope of the study is limited and focuses only on the acceleration of polynomial multiplication in the encryption process of the NTRU Round-3 version. The study does not address other potential vulnerabilities or attack vectors in the NTRU cryptosystem or other Post-Quantum Cryptography algorithms. Additionally, the evaluation is limited to a specific parameter set and hardware platform, which may not generalize to other scenarios.
6. It is difficult for individuals who are unfamiliar with the subject to grasp since the author believes that readers are knowledgeable with the issue. The author touches on several significant topics, but some of the more complex concepts lack explanation or clarity, which may make it challenging for readers to grasp the article's overall context.
7. The proposed solution seems promising, providing more detailed and comprehensive information could help strengthen the work and make it more impactful. Also, addressing the weakness mentioned in the comments can also help improving the paper.
8. The figures are visually appealing and of high quality. Additionally, the captions are descriptive. The evaluation results cover all details outlined in the proposed solution, and tables were used to present the data. On the whole, the result section offers a clear and detailed depiction of the evaluation outcomes.
9. The NTRU cryptosystem is employed by the authors to implement a hardware/software co-design method to speed up polynomial multiplication in the encryption process for devices with limited resources. In accordance with the software-level multiplication runtime, the developed multiplier reportedly increased speed by up to 30-45 times with a device occupancy of about 5%. Although the research doesn't specifically compare its results with those of other studies, the increased speed and resource usage may be seen as a major improvement over the software solution that is currently in use. The study offers useful information on the creation of hardware accelerators for polynomial multiplication in cryptographic algorithms, particularly for devices with limited resources.
10. Overall, the paper proposes a hardware dedicated architecture to accelerate the polynomial multiplication in the encryption process of the NTRU version presented in the third round of the NIST contest. The proposed solution contemplates a reduction in the number of clock cycles by considering the zero coefficients of the obfuscation polynomial r(x), but a new method is proposed that allows speeding up the multiplication operation without compromising the security of the cryptosystem. The paper also provides implementation results and an analysis in terms of efficiency, which shows that it is possible to accelerate the NTRU Round-3 encryption scheme between 30 and 40 times with respect to the reference software version with a resource consumption of 8% approximately.
Round 2
Reviewer 1 Report
In this revision, the authors have addressed my concerns, I suggest it be accepted.
Reviewer 2 Report
Dear authors,
I have reviewed the new version of his document and I considered that the comments of the previous version were resolved, in terms of table reduction, and correction of various phrases, as well as an adequate description of the keywords.
Greetings, and success in your projects.